# Temporal trends in relative survival following percutaneous coronary intervention

William J Hulme,[1] Matthew Sperrin,[1] Glen Philip Martin,[1] Nick Curzen,[2] Peter Ludman,[3] Evangelos Kontopantelis,[1] Mamas A Mamas,[4,5] on behalf of the British Cardiovascular Intervention Society and the National Institute of Cardiovascular Outcomes Research

For numbered affiliations see end of article.

**Correspondence to**
Dr William J Hulme;
william.hulme@manchester.ac.uk

## ABSTRACT

**Objective** Percutaneous coronary intervention (PCI) has seen substantial shifts in patient selection in recent years that have increased baseline patient mortality risk. It is unclear to what extent observed changes in mortality are attributable to background mortality risk or the indication and selection for PCI itself. PCI-attributable mortality can be estimated using relative survival, which adjusts observed mortality by that seen in a matched control population. We report relative survival ratios and compare these across different time periods.

**Methods** National Health Service PCI activity in England and Wales from 2007 to 2014 is considered using data from the British Cardiovascular Intervention Society PCI Registry. Background mortality is as reported in Office for National Statistics life tables. Relative survival ratios up to 1 year are estimated, matching on patient age, sex and procedure date. Estimates are stratified by indication for PCI, sex and procedure date.

**Results** 549 305 procedures were studied after exclusions for missing age, sex, indication and mortality status. Comparing from 2007 to 2008 to 2013–2014, differences in crude survival at 1 year were consistently lower in later years across all strata. For relative survival, these differences remained but were smaller, suggesting poorer survival in later years is partly due to demographic characteristics. Relative survival was higher in older patients.

**Conclusions** Changes in patient demographics account for some but not all of the crude survival changes seen during the study period. Relative survival is an under-used methodology in interventional settings like PCI and should be considered wherever survival is compared between populations with different demographic characteristics, such as between countries or time periods.

### Strengths and limitations of this study

► Using national registry data, this study is the first to compare patient survival after percutaneous coronary intervention with matched population survival across an entire healthcare system.
► It provides a template for similar investigations in other interventional settings where relative survival is under-used.
► Demographic matching in addition to age, sex and year of procedure was not possible, though life tables stratified by other characteristics may be available from the Office for National Statistics in future.

of acute coronary syndrome (ACS) cases increasing from 53.6% to 64.3%[9] in the same period. In the USA, data from the CathPCI registry indicate an increase in the proportion of PCIs for ACS from 57.0% to 64.3% from 2011 to 2014.[3 10]

Correspondingly, patient case mix has also changed such that PCI is now more likely to be performed in older, more comorbid patients at higher risk of early mortality. For instance, in the UK between 2007 and 2015, mean patient age increased from 63.6 to 65.1 years and the proportion of patients with diabetes rose from 17.5% to 22.0%.[9] This is due, in part, to more permissive patient selection criteria and a move to more emergent indications where there is less scope for case selection and increased access,[11 12] though changes in the characteristics of the general population may partially explain this transition. Indeed, improvements in secondary prevention for cardiovascular disease and wider societal changes in general may be a significant driver of changes to baseline patient risk given a persistent, although slowing, decline in age-standardised cardiovascular mortality in both the UK[13] and the USA.[14] In addition, the evidence-base

## INTRODUCTION

The worldwide use of percutaneous coronary intervention (PCI) for coronary revascularisation has expanded in recent years,[1–6] though differences in this change have been observed internationally.[7 8] In the UK, the number of procedures increased from around 78 000 in 2007 to 97 000 in 2016, with the proportion

concerning optimal intervention techniques and treatment strategies has evolved, prompting changes to procedural practice towards the treatment of more complex disease.[15–17]

Mortality rates following PCI will be affected by these trends, though it is unclear to what extent the any changes are related to the baseline mortality risk of selected patients or procedural practice. Consequently, there is a need to report mortality rates from national PCI registries within the context of the changing PCI patient demographic, relative to corresponding changes in the background population.

Relative survival is a statistical technique that compares the survival of a disease or treatment group with the survival of the general population matched on one or more characteristics, such as age and sex, for example as presented in national life-expectancy tables.[18 19] The excess mortality—that is, the mortality after adjusting for the expected mortality estimated from the control population— associated with that group can then be quantified. Relative survival methods are typically used to study survival differences following disease diagnosis or to compare long-term treatment strategies where excess mortality develops slowly, for example in cancer.[20 21] More recently, it is being considered in cardiovascular settings,[22–24] including PCI,[25–29] though these have focused on long-term survival and did not consider changes over time.

We aimed to compare mortality rates following PCI with age-matched and sex-matched mortality rates found nationally and to examine if these comparisons differ between early or contemporary cohorts.

## METHODS

### PCI procedure data
The British Cardiovascular Intervention Society (BCIS) audits all PCI activity in the UK, with data collection managed by the National Institute for Cardiovascular Outcomes Research (NICOR). Information on each procedure is captured locally by hospitals and uploaded to a central server. Mortality tracking is available via the Office for National Statistics (ONS) for procedures in England and Wales for patients with a valid National Health Service number. Mortality tracking was available up to May 2015.

Procedures in England and Wales between 2007 and 2014 for patients aged between 18 and 100 years old were extracted. Procedures with missing age, sex, indication and mortality status were excluded.

### ONS life-table data
Expected mortality rates for England and Wales are available from ONS national life-tables stratified by age in 1 year bands, se, and calendar year.[30]

### Statistical analysis
Each procedure was matched to expected mortality life-table data based on the patients' age, sex and the date of the procedure. Survival models were stratified by procedure date (2007–2008, 2009–2010, 2011–2012, 2013–2014), and by patient sex and indication (elective, unstable angina or non-ST-elevation myocardial infarction (UA/NSTEMI), ST-elevation myocardial infarction (STEMI)) as these are known determinants of survival, and there are distinct survival differences between these groups. Two-year time periods were chosen to ensure temporal trends are sufficiently captured while maintaining clarity of exposition. Analyses consider mortality up to 1 year postprocedure with mortality censored thereafter, as long-term estimates are confounded by the increasing influence of competing contributors to mortality and time-varying risk factors.[31]

Crude survival rates are plotted, with survival differences between periods compared using the log-rank test.[32 33] Relative survival models are built that adjust for expected survival using the Ederer II method.[18 19] Relative survival estimates are plotted, and differences between periods compared using the log-rank test after a transformation of survival time described by Stare *et al.*[34] The observed (PCI) and expected (life-table) 1-year mortality rates are also compared by considering their ratios across different ages. This expresses the relationship of relative survival and age without explicitly modelling the hazard function, which is challenging in this setting since the hazard falls extremely steeply in the early stages of follow-up. This quantity is plotted, using general additive models with a binomial link to smooth observed mortality.

The relative survival estimate, calculable at each day postprocedure, is the ratio of the observed survival proportion of those who had PCI to the expected survival proportion based on a matched population. This quantity can interpreted as the post-PCI survival rate if patients were only exposed to PCI-related mortality risk and not other background risks. PCI-related mortality risk should be understood as the mortality risk of being a person who had PCI, not simply the risk of the procedure itself. If the relative survival estimate levels-off after a given time then, on average, there is no additional PCI-related mortality risk for those who survived up to that time. If the relative survival estimate increases then, on average, mortality was less than expected had the patients not needed PCI. In most interventional settings, this quantity is typically less than one, indicating that PCI is associated with a greater mortality risk than no PCI.

All analyses were performed using R V.3.4.4.[35] The tidyverse data manipulation and visualisation suite[36] was used throughout. The relsurv package[37–39] was used for relative survival modelling. The complete R script is available on GitHub,[40] with synthetic BCIS data provided to replicate analysis steps without disclosing proprietary BCIS data. Access to study materials was only required for the first author.

### Patient involvement
This is a registry-based study with all data collected prior to the design phase and outcomes necessarily restricted

**Table 1** Number of procedures by period, sex, indication

| | 2007–2008 | | 2009–2010 | | 2011–2012 | | 2013–2014 | | 2007–2014 | |
|---|---|---|---|---|---|---|---|---|---|---|
| | N | % | N | % | N | % | N | % | N | % |
| Female | | | | | | | | | | |
| Elective | 13 847 | 45.2 | 13 392 | 38.4 | 12 603 | 33.0 | 12 093 | 31.3 | 51 935 | 36.5 |
| UA/NSTEMI | 12 756 | 41.6 | 13 858 | 39.8 | 15 139 | 39.6 | 15 594 | 40.4 | 57 347 | 40.3 |
| STEMI | 4043 | 13.2 | 7608 | 21.8 | 10 480 | 27.4 | 10 908 | 28.3 | 33 039 | 23.2 |
| Total | 30 646 | | 34 858 | | 38 222 | | 38 595 | | 142 321 | |
| Male | | | | | | | | | | |
| Elective | 40 246 | 46.3 | 40 284 | 40.4 | 39 715 | 36.3 | 39 228 | 35.3 | 159 473 | 39.2 |
| UA/NSTEMI | 33 797 | 38.9 | 37 049 | 37.2 | 39 379 | 36.0 | 40 749 | 36.7 | 150 974 | 37.1 |
| STEMI | 12 921 | 14.9 | 22 306 | 22.4 | 30 244 | 27.7 | 31 066 | 28.0 | 96 537 | 23.7 |
| Total | 86 964 | | 99 639 | | 109 338 | | 111 043 | | 406 984 | |
| Male and female | | | | | | | | | | |
| Elective | 54 093 | 46.0 | 53 676 | 39.9 | 52 318 | 35.5 | 51 321 | 34.3 | 211 408 | 38.5 |
| UA/NSTEMI | 46 505 | 39.6 | 50 745 | 37.8 | 54 314 | 36.9 | 56 152 | 37.7 | 208 321 | 37.9 |
| STEMI | 16 964 | 14.4 | 29 914 | 22.2 | 40 724 | 27.6 | 41 974 | 28.1 | 129 576 | 23.6 |
| Total | 117 610 | | 134 497 | | 147 560 | | 149 638 | | 549 305 | |

to mortality only. Therefore, patients were not involved in the design of this study.

## RESULTS

In total, there were 575 203 procedures recorded in the BCIS-NICOR registry between 2007 and 2014 in England and Wales, of which 549 305 (95.5%) were available for analysis after exclusions due to missing data (see online supplementary table A1). Follow-up is complete for every procedure up to 1 year (ie, no censoring occurred before 1 year), except for procedures from June to December 2014, where available follow-up time was less than 1 year.

Table 1 presents procedure numbers by year, sex and indication, and online supplementary table A2a-c present baseline patient characteristics and mortality rates for elective, UA/NSTEMI and STEMI procedures, respectively. Female patients made up a quarter (25.9%) of procedures. Mean patient age increased from 63.8 years old in 2007–2008 to 65.1 years old in 2013–2014, with similar increases observed within indications (64.5 to 65.8 years for elective procedures; 63.6 to 65.7 years for UA/NSTEMI; 62.1 to 63.5 years for STEMI). PCI for ACS (UA/NSTEMI/STEMI) accounted for 61.5% of procedures though this proportion increased from 54.0% to 65.7%.

Figure 1 presents the observed survival rates without adjustment for expected survival. Across all indications mortality risk is highest immediately following the procedures as this is where the survival curve gradient is steepest. The log-rank test indicates some differences in crude survival rates between periods and, typically, more recent years are associated with lower survival. Survival differences by indication and sex are also evident, with survival consistently lower in women. Comparing crude

survival from 2007 to 2008 to 2013–2014 at 1 year: elective procedures, 97.7% to 97.5% for women and 98.0% vs 97.7% for men; UA/STEMI, 94.6% vs 93.7% for women and 95.8% vs 94.4% for men; STEMI, 87.5% vs 86.4% for

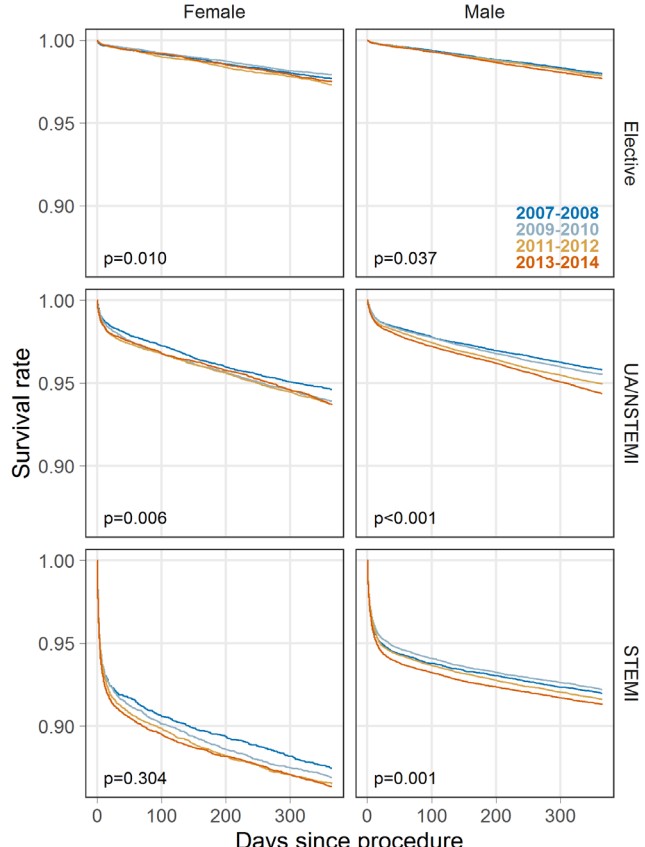

**Figure 1** Observed survival rates by patient indication and year of procedure. P values are from the log-rank test.

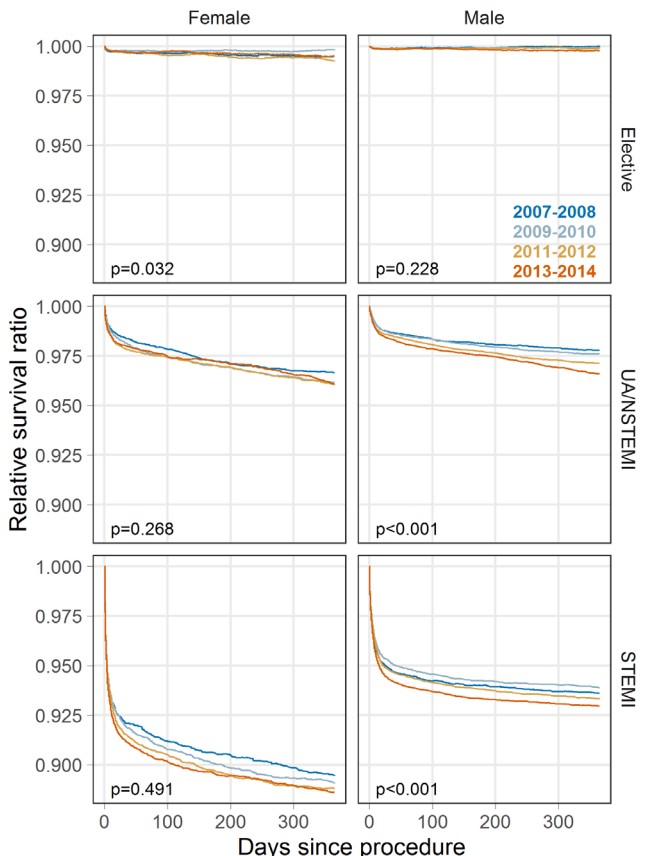

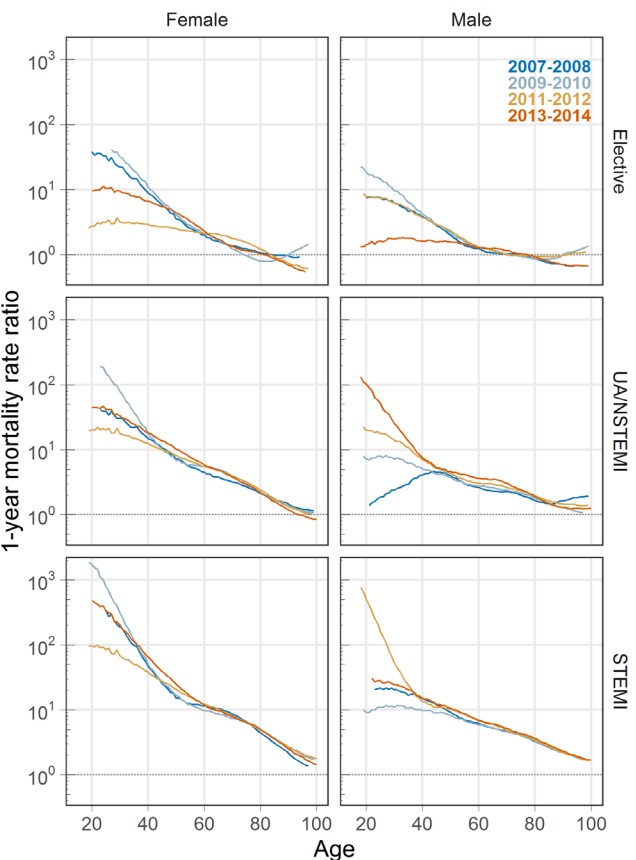

**Figure 2** Relative survival estimates by patient indication and year of procedure. P values are from a log-rank-type test for relative survival curves.

**Figure 3** Observed-to-expected 1-year mortality ratio (log-scale) by patient age. Binomial general additive models with 7-knot splines are used for smoothing the observed 1-year mortality, then divided by the expected mortality rate. 95% confidence limits are removed for clarity.

women and 92.0% vs 91.3% for men (see online supplementary table A2).

Figure 2 shows the relative survival estimate. For elective procedures relative survival levels-off quickly, indicating that procedural mortality risk is present immediately following the procedure but that on average, risk returns to the matched population baseline. For ACS procedures, relative survival continues to decline up to 1-year postprocedure, indicating long-term excess risk in these patients compared with the matched population. The log-rank test for relative survival differences reveals smaller differences than for crude survival, though there remains strong evidence of survival differences for men following PCI for UA/STEMI or STEMI. Comparing relative survival from 2007 to 2008 to 2013–2014 at 1 year: elective procedures, 99.5% to 99.5% for women and 100% vs 99.8% for men; UA/STEMI, 96.7% vs 96.1% for women and 97.8% vs 96.6% for men; STEMI, 89.5% vs 88.6% for women and 93.6% vs 93.0% for men (see online supplementary table A3).

Figure 3 shows the ratio between observed and expected 1 year mortality rates by patient age. For younger patients, observed mortality is considerably higher than expected mortality, but this gap reduces with patient age. For instance, observed mortality at 1 year following elective PCI is 2–5 times greater than expected at age 50, but

these are equal by age 80. For UA/NSTEMI, this ratio is between 5 and 10 at age 50, though is less than 3 by age 80. For STEMI, this ratio decreases from around 9–15 to around 5 from age 50 to 80. There are no clear trends over time.

Confidence intervals at the 95% level for figures 1–3 are provided in supplementary materials (online supplementary figure A1-3), as are plots of the cumulative excess hazard rate (online supplementary figure A4), which may be interpreted as the total additional exposure to mortality risk at any point in time for patients with PCI compared with the general population.

## DISCUSSION

We present results for 549 305 PCI procedures over an 8-year period showing observed, expected and relative survival estimates over time. We demonstrate that, after adjustment for patient age, sex and procedure date using national life-table mortality rates, differences in patient survival over time are present in men following PCI for ACS, though these differences are small (relative survival at 1 year: UA/NSTEMI 97.8% in 2007–2008 vs 96.6% in 2013–2014; STEMI 93.6% in 2007–2008 vs 93.0% in

2013–2014). Small, non-significant relative survival differences were found within other strata. However, the potential counteracting effects of selection factors (that may have increased mortality risk) and of improvements in processes of care (that may have reduced mortality risk) cannot be ruled out.

For elective cases, there is evidence that relative survival increases after the initial postprocedural risk. This suggests that, conditional on surviving the initial risk shock, mortality is on average lower than in the matched population. This is likely driven by patient selection factors, particularly for more elderly patients, where there may be selection bias such that PCI is undertaken in only the healthiest patients. Another factor may be that these patients are highly likely to be supported with secondary prevention treatments for coronary artery disease, whereas in the general population, there may be many people with undetected or untreated coronary artery disease (CAD).

Mortality increases with age at a rate that is lower in the PCI population than in an equivalent matched population. In other words, though mortality is naturally higher as age increases, older patients treated by PCI have a mortality rate more similar to the rate in the general population than younger patients. This is true across all indications. In general then, relative survival is higher for older patients compared with younger patients, though in absolute terms their prognosis is still poorer.

This may partly be determined by clinician selection practices, such that younger patients with CAD are more likely to be referred for PCI than older patients for whom PCI is not optimal. For instance, any 80-year-old considered well enough to undergo elective PCI is likely to have fewer comorbidities and frailties compared with other 80-year-old on average. Equally, epidemiological selection suggests that a 40-year-old requiring elective PCI will undoubtedly have a higher baseline risk for mortality than the average 40-year-old since the prevalence of cardiovascular disease is low at this age. For urgent or emergency procedures, simply being part of the PCI registry suggests a degree of patient resilience having experienced an acute, life-threatening event yet survived long enough to be treated. At the onset of an ACS, as older patients are more likely than younger patients to die before receiving PCI or any other treatment, those older patients that do receive PCI will typically be less frail than others of a similar age. The distinct trend for higher relative survival in older patients is thus clear evidence of these selection processes at play, and demonstrates the importance of accounting for these imbalances when analysing registry data.

Across all indications, both absolute and relative survival is lower in women than in men. This is a well-known phenomenon in cardiovascular settings variously attributed to differences in age, comorbidities, time to treatment and secondary prevention.[41–44] This work demonstrates unequivocally that differences in age and background mortality rates do not sufficiently account for this disparity.

In general, both crude and relative survival estimates are lower in recent years. Assuming a consistent data collection methodology over time, realistically this can either be attributed to an average reduction in procedural efficacy or an increase in higher risk procedures that is not solely explained by age-adjusted and sex-adjusted background mortality. In the context of rapid changes to PCI patient case mix and practice in recent years,[45] including more permissive patient selection criteria and increasing use of PCI in favour of alternative revascularisation strategies, the small decline in relative survival observed from 2007 to 2014 is unsurprising. Further adjustment for patient case mix will demonstrate if and where actual survival gains have been made, or indeed if survival in certain groups of patients has not improved over time.

Relative survival offers distinct advantages over crude survival rates.[19] The relative mortality estimate (one minus relative survival) may be interpreted as an approximation of the excess mortality attributable jointly to the indication for intervention and the intervention itself. This allows PCI-specific mortality to be measured without needing cause of death information which may often be inconsistently recorded or unavailable completely. Relative survival also conveniently accounts for mortality that may be only indirectly related to PCI, for instance non-cardiac mortality secondary to postoperative bleeding.

As the underlying population may vary considerably across different countries, intervention registries or time periods, methods that appropriately adjust for background population mortality provide more suitable survival estimates for comparison in time and space than crude survival alone, provided the comparisons are valid.[46 47] However, despite the popularity of relative survival methodology to compare survival in diseased versus non-diseased populations, in particular cancer, these methods have only recently seen some use in interventional settings like PCI, and relative survival is not routinely reported by PCI registries. This study concisely demonstrates the feasibility and appropriateness of the relative survival approach in such settings and provides a useful template for producing population-adjusted PCI survival estimates in the future.

### Limitations
Patients are matched to national mortality estimates by year, age and sex only. Despite stratification of models by these factors, substantial case heterogeneity will remain that is not modelled. No account is taken for other potentially unbalanced factors such as ethnicity, socialeconomic status and comorbidities. Accounting for these factors is not possible with the simple relative survival approach used here as national life-tables stratifying by these factors are not readily available. However, using regression modelling in this setting is challenging as the mortality risk following PCI is raised acutely and the hazard rate changes rapidly; modelling the hazard must be done with

extreme care as many assumptions of standard regression techniques are unlikely to hold.

Patients will be present more than once in the PCI cohorts if they have been readmitted for subsequent PCI within the study period. The proportion of procedures in patients already present in the cohort is 13.5%, though this is reduced to 7.3% when considering indications separately.

Life-table data is based on national mortality statistics and therefore is not strictly independent of the PCI registry, since all patients will contribute to population and mortality statistics. This is an unavoidable complication in the absence of stratification by CAD or PCI status that biases results towards no relative survival difference. However, given the analysis cohort constitutes less than 1% of the population of England and Wales, this bias can be considered negligible.

As the log-rank tests stratify groups independently it will ignore the inherent correlation between adjacent or near-adjacent years. It does not provide point estimates of differences between strata as all differences through time contribute to the test statistic. It is therefore most useful when used alongside a graphical representation of survival differences.

## CONCLUSION

We demonstrate that the decline in indication-stratified crude survival rate over time is less apparent after adjusting for expected population mortality matched on age, sex and procedure year, though patient demographics do not explain all of the difference. Adjusting for additional patient risk factors is necessary to identify other drivers of mortality change.

**Author affiliations**
[1]Farr Institute, Faculty of Biology, Medicine and Health, University of Manchester, Manchester Academic Health Science Centre, Manchester, UK
[2]University Hospital Southampton and Faculty of Medicine, University of Southampton, Southampton, UK
[3]Department of Cardiology, Queen Elizabeth Hospital, Birmingham, UK
[4]Keele Cardiovascular Research Group, Centre for Prognosis Research, Institute of Primary Care and Health Sciences, University of Keele, Keele, UK
[5]Academic Department of Cardiology, Royal Stoke Hospital, Stoke-on-Trent, UK

**Collaborators** British Cardiovascular Intervention Society; National Institute for Cardiovascular Outcomes Research

**Contributors** WJH, MAM: conceived the study. WJH, GPM, MS, EK: participated in the design of the study. WJH: performed the statistical analysis and drafted the manuscript. WJH, GPM, MS, EK, NC, PL, MAM: interpreted the data and results, made important contributions and revisions to the work and read and approved the final manuscript.

**Funding** This work was supported by MRC grant number MR/K006665/1.

**Competing interests** None declared.

**Patient consent for publication** Not required.

**Provenance and peer review** Not commissioned; externally peer reviewed.

**Data sharing statement** Analysis scripts and a synthetic test dataset are available on GitHub: https://zenodo.org/badge/latestdoi/131696652

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
