## [Reviewer comments · BMJ Open]

ARTICLE DETAILS

TITLE (PROVISIONAL)	Temporal trends in Relative Survival Following Percutaneous Coronary Intervention
AUTHORS	Hulme, William; Sperrin, Matthew; Martin, Glen; Curzen, Nick; Ludman, Peter; Kontopantelis, Evangelos; Mamas, Mamas

VERSION 1 – REVIEW

REVIEWER	Varunsiri Atti Michigan State University, USA
REVIEW RETURNED	04-Aug-2018

GENERAL COMMENTS	Hulme et al report the relative survival rates after PCI using national database of the U.K. The authors made a very nice job calculating the relative survival rates. I think this has not been reported in prior literature to my knowledge. Few comments to the author are Page 7, authors mention figures as figure 1, 2, 3. However the figures at end of manuscript were named A1, A2, A3. Page 9 is empty. Page 21, line 23. The sentence is not clear about what exactly it is communicating there. Can you please correct or explain Page 10, paragraph 4. The writing is little generalized. Would recommend some modification. Page 11, line 48. Recommend editing the word neatly.
--

REVIEWER	HiroYuki Daida Juntendo University, Japan
REVIEW RETURNED	01-Nov-2018

GENERAL COMMENTS	We have read with interest the article written by Hulme W et al entitled "Temporal trends in relative survival following percutaneous coronary intervention". In this article, the authors evaluated the mortality rates following PCI matching on age and sex, and procedure date using British PCI registry data from 2007 to 2014. They also demonstrated comparison of observed (PCI) mortality and expected (life-table) one-year mortality based on national mortality statistic. There are some issues which should be resolved and clarified to improve the article. Major comments 1. As the authors mentioned, the main limitation of the present study was that patients are adjusted only by age, sex and PCI date. Although it may be difficult to analyze using regression model, adjusting confounders are necessary for discussion. What
--

	we would like to know is the time-dependent change of prognostic factors in accordance with the improvement of standard care including coronary intervention and medical therapy over time. Please ask statisticians whether to evaluate the adequate method identifying the time-dependent change of predictor for mortality. Minor comments 1. Tables were relatively complicated to understand because of too much amount of displayed information.
--	--

REVIEWER	Abdul Mozid Bristol Heart Institute, UK
REVIEW RETURNED	12-Nov-2018

GENERAL COMMENTS	The authors present retrospective analysis of the UK BCIS PCI registry linked to mortality data from the Office for National Statistics. The aim of this study was to assess the use of relative survival (using matched controls from ONS life-tables) as a method to compare temporal changes in mortality from PCI. Comparing 2007-2008 to 2013-14, 1 year crude survival was consistently lower in later years across all groups. For relative survival, the differences remained but were smaller, suggesting poorer survival in later years is only partly due to demographic characteristics. Relative survival was higher in older patients. This is a very well written manuscript assessing a very relevant research topic as there has been steady growth in PCI nationally with more elderly and complex patients being treated. Relative survival seems a useful tool to compare PCI outcome over time with a changing demographic. In the manuscript relative survival estimates are stratified by indication for PCI, sex and procedure date. Would it be possible to stratify further according to case complexity? Has increasing complexity of cases contributed to the small decline in survival?, i.e. is there a relationship between number of CTOs, LMS, MVD, rotablation procedures and change in relative survival. This data should be available within the current BCIS dataset. There continues to be a decline in rate of CABG and it is likely patients are instead being offered complex PCI with lower threshold for surgical decline. Otherwise, excellent paper with no other suggestions.
--

VERSION 1 – AUTHOR RESPONSE

Reviewer: 1

Reviewer Name: Varunsiri Atti

Institution and Country: Michigan State University, USA

Please state any competing interests or state 'None declared': None declared

Hulme et al report the relative survival rates after PCI using national database of the U.K. The authors made a very nice job calculating the relative survival rates. I think this has not been reported in prior literature to my knowledge. Few comments to the author are:

Page 7, authors mention figures as figure 1, 2, 3. However the figures at end of manuscript were named A1, A2, A3.

Figures A1, A2, and A3 are in supplementary materials, placed at the end of the document when the originally submitted documents are collated and converted to pdf. Figures 1, 2, and 3 are available to view on pages 22-24 of the collated pdf document.

Page 9 is empty.

Unfortunately authors have no control over the conversion of original submission documents to the final pdf version for reviewers. Document is correctly formatted in the original submission. I expect this will be resolved by copy editors if this manuscript is accepted for publication.

Page 21, line 23. The sentence is not clear about what exactly it is communicating there.

There is no line 23 on page 21. Page 21 describes figure legends.

Can you please correct or explain Page 10, paragraph 4. The writing is little generalized. Would recommend some modification.

Thank you for this suggestion, we agree the text is rather laboured and have made the following changes-

Original text:

This may partly be determined by clinician selection practices, such that younger patients with CAD are more likely to be referred for PCI than older patients for whom PCI is not optimal. For instance, for elective procedures, any 80-year-old considered well enough to undergo elective PCI is likely to be otherwise healthy and without the typical contraindications for PCI such as intolerance of antiplatelet therapies or other significant co-morbidities and frailties. Conversely, a 40-year-old requiring elective PCI clearly has a higher baseline risk for mortality than the average 40-year-old. However, epidemiological selection processes are also relevant. Irrespective of the decision to treat CAD with PCI, younger PCI patients are sicker than older PCI patients relative to a matched population, given that competing mortality risks loom larger in an older population where death from other causes such as cancer or respiratory illnesses is more likely. Finally, considering urgent or emergency procedures, simply being part of the PCI registry suggests a degree of patient resilience having experienced an acute event but survived long enough to be treated. Due to increasing frailty, older patients should expect inferior odds than younger patients of receiving PCI before dying, so those older patients that do survive to PCI will typically be less frail than others of a similar age. The distinct trend for higher relative survival in older patients is thus clear evidence of these selection processes at play, and the importance of accounting for these imbalances when analysing registry data.

Revised text:

This may partly be determined by clinician selection practices, such that younger patients with CAD are more likely to be referred for PCI than older patients for whom PCI is not optimal. For instance, any 80-year-old considered well enough to undergo elective PCI is likely to have fewer co-morbidities and frailties compared with other 80-year-olds on average. Equally, epidemiological selection suggests that a 40-year-old requiring elective PCI will undoubtedly have a higher baseline risk for mortality than the average 40-year-old since the prevalence of cardiovascular disease is low at this age. For urgent or emergency procedures, simply being part of the PCI registry suggests a degree of patient resilience having experienced an acute, life-threatening event yet survived long enough to be treated.. At the onset of an ACS, as older patients are more likely than younger patients to die before receiving PCI or any other treatment, those older patients that do receive PCI will typically be less frail than others of a similar age. The distinct trend for higher relative survival in older patients is thus clear evidence of these selection processes at play, and demonstrates the importance of accounting for these imbalances when analysing registry data.

Page 11, line 48. Recommend editing the word neatly

Original text:

Relative survival also neatly accounts for mortality that may be only indirectly related to PCI, for instance non-cardiac mortality secondary to post-operative bleeding.

Revised text:

Relative survival also conveniently accounts for mortality that may be only indirectly related to PCI, for instance non-cardiac mortality secondary to post-operative bleeding.

Reviewer: 2

Reviewer Name: Hiroyuki Daida

Institution and Country: Juntendo University, Japan

Please state any competing interests or state 'None declared': None declared

We have read with interest the article written by Hulme W et al entitled "Temporal trends in relative survival following percutaneous coronary intervention". In this article, the authors evaluated the mortality rates following PCI matching on age and sex, and procedure date using British PCI registry data from 2007 to 2014. They also demonstrated comparison of observed (PCI) mortality and expected (life-table) one-year mortality based on national mortality statistic. There are some issues which should be resolved and clarified to improve the article.

Major comments

1. As the authors mentioned, the main limitation of the present study was that patients are adjusted only by age, sex and PCI date. Although it may be difficult to analyze using regression model, adjusting confounders are necessary for discussion. What we would like to know is the time-dependent change of prognostic factors in accordance with the improvement of standard care including coronary intervention and medical therapy over time. Please ask statisticians whether to evaluate the adequate method identifying the time-dependent change of predictor for mortality.

One of the major motivations of this article was to demonstrate that simple, minimal-assumption relative survival methods can be used to communicate outcome differences whilst appropriately adjusting for age and sex, *without* needing to rely on the assumptions and carefully calibrated parameterisations that are necessary for regression modelling. In the discussion we advocate for the use of relative survival methods for routine reporting of mortality outcomes over time in large treatment registries, especially where demographics are shifting. This allows for background survival to be adjusted for without needing to resort to stratification or parametric modelling. So, while we acknowledge this is a limitation for understanding the possible confounding effects of other clinical characteristics, this is not the focus of this study. Using regression models would lead to a different study design and message altogether.

What's more, for high-risk post-procedural outcomes, parameterisation of the hazard (or cumulative hazard) is rarely an effective strategy because the hazard is initially so high and decreases so rapidly. Even generalised-F models (a highly-flexible, 4-parameter function that includes as special cases the Weibull, the generalised-Gamma, the log-normal, and the log-logistic survival distributions) cannot adequately account for these highly volatile hazards. A more sophisticated approach would be needed, such as Royston-Parmar. Again, this is beyond the scope of this study and detracts from the simplicity that relative survival methods bring to mortality differentiation following PCI.

Minor comments

1. Tables were relatively complicated to understand because of too much amount of displayed information.

We agree the tables are occasionally quite busy but these were included for reference and for the more curious reader. They are not the main focus of this study. If a reader has a particular interest in, for example, the distribution of ethnicities in a particular sex-indication strata, then the information is available to find in these tables. This is a case of not being able to please everybody and, in such a scenario, rather than split the difference we believe it's better to provide more rather than less information.

Reviewer: 3

Reviewer Name: Abdul Mozid

Institution and Country: Bristol Heart Institute, UK

Please state any competing interests or state 'None declared': None declared

The authors present retrospective analysis of the UK BCIS PCI registry linked to mortality data from the Office for National Statistics. The aim of this study was to assess the use of relative survival (using matched controls from ONS life-tables) as a method to compare temporal changes in mortality from PCI. Comparing 2007-2008 to 2013-14, 1 year crude survival was consistently lower in later years across all groups. For relative survival, the differences remained but were smaller, suggesting poorer survival in later years is only partly due to demographic characteristics. Relative survival was higher in older patients.

This is a very well written manuscript assessing a very relevant research topic as there has been steady growth in PCI nationally with more elderly and complex patients being treated. Relative survival seems a useful tool to compare PCI outcome over time with a changing demographic.

In the manuscript relative survival estimates are stratified by indication for PCI, sex and procedure date. Would it be possible to stratify further according to case complexity? Has increasing complexity of cases contributed to the small decline in survival?, i.e. is there a relationship between number of CTOs, LMS, MVD, rotablation procedures and change in relative survival. This data should be available within the current BCIS dataset. There continues to be a decline in rate of CABG and it is likely patients are instead being offered complex PCI with lower threshold for surgical decline.

The specific focus of this article was a) a broad investigation in recent mortality trends in PCI b) to see what relative survival offers in addition to crude survival alone, and c) to argue that relative survival should be considered more readily in settings where mortality trends are routinely reported, and so we considered that the inclusion of additional factors was beyond its scope.

Further, there were two practical issues that prevented us from taking this approach ourselves.

- 1) Data completeness for our stratification variables - age, sex, and indication- are near-perfect, and can be confidently dealt with using a complete-case analysis, as was done here. However, for CTOs, LMS stenosis, MVD, data are less complete and a complete-case analysis is not as reliable (data on pre-PCI stenosis in the coronary arteries is reported between 60-80% of the time, with improved completeness in more recent years). In regression models, missing data can typically be dealt with using multiple imputation, under certain assumptions. Multiple imputation provides unbiased estimates of model parameters (eg odds-ratios) with standard errors that reflect the uncertainty due to imputation (rather than simply 'filling in the blanks' for individual patients – a common misconception). However, in the present analysis there are no such model parameters to estimate - only survival curves. While point-estimates of Kaplan-Meier survival curves are easily obtained with multiple imputation, obtaining standard errors is slightly trickier. Either way, since this paper attempts

to demonstrate the ease with which relative survival methods can be employed in the routine reporting of mortality, multiple imputation was deliberately avoided.

- 2) There is a limit to the number of stratifying variables we can reasonably consider in a single article, since each additional variable increases the number of strata multiplicatively. Currently we have 2 sexes * 3 indications * 4 two-year periods = 24 strata. Adding CTO for example would create 48 strata in total if we are interested in trends in CTO whilst also accounting for age, indication, and sex. This is a clear limitation of non-regression-based approaches but, as outlined in comments above, this was not the focus of our study.

Having said that, we agree that stratification by additional factors relating to the cardiovascular condition of the patient at admission would be interesting and certainly worth considering in future work.

Otherwise, excellent paper with no other suggestions.

VERSION 2 – REVIEW

REVIEWER	Abdul Mozid Bristol Heart Institute
REVIEW RETURNED	19-Nov-2018
GENERAL COMMENTS	Thanks for clarifying